# Innate immune control of influenza virus interspecies adaptation via IFITM3

Parker J. Denz [1,2], Samuel Speaks[1,2], Adam D. Kenney[1,2], Adrian C. Eddy[1,2], Jonathan L. Papa[1,2], Jack Roettger [1,2], Sydney C. Scace[1,2], Adam Rubrum[3], Emily A. Hemann[1,2], Adriana Forero [1,2], Richard J. Webby [3], Andrew S. Bowman [2,4] & Jacob S. Yount [1,2] ✉

Influenza virus pandemics are caused by viruses from animal reservoirs that adapt to efficiently infect and replicate in human hosts. Here, we investigate whether Interferon-Induced Transmembrane Protein 3 (IFITM3), a host anti-viral factor with known human deficiencies, plays a role in interspecies virus infection and adaptation. We find that IFITM3-deficient mice and human cells can be infected with low doses of avian influenza viruses that fail to infect WT counterparts, identifying a new role for IFITM3 in controlling the minimum infectious virus dose threshold. Remarkably, influenza viruses passaged through *Ifitm3*[−/−] mice exhibit enhanced host adaptation, a result that is distinct from viruses passaged in mice deficient for interferon signaling, which exhibit attenuation. Our data demonstrate that IFITM3 deficiency uniquely facilitates potentially zoonotic influenza virus infections and subsequent adaptation, implicating IFITM3 deficiencies in the human population as a vulnerability for emergence of new pandemic viruses.

Respiratory virus emergence from animal reservoirs into the human population is a continuous worldwide health threat. The influenza pandemics of 1918, 1957, 1968, and 2009 arose from viruses of avian or swine origin that adapted to replicate robustly in humans[1,2]. Likewise, infection of humans with coronaviruses, such as MERS-CoV, SARS-CoV, and SARS-CoV-2, likely resulted from animal-to-human transmission events followed by virus adaptation to the human respiratory tract and other niches[3]. Despite these notable examples, major outbreaks of new viruses could be considered rare given the routine contact between humans and wildlife throughout the world[4]. This discrepancy suggests that natural impediments to interspecies virus transmission or adaptation exist in human cells. Conversely, genetic defects in such host defense mechanisms could provide opportunities for viruses to enter and more efficiently adapt to humans. Here, we tested the hypothesis that deficiencies in the innate immunity protein, Interferon-Induced Transmembrane Protein 3 (IFITM3), facilitate interspecies infection and adaptation by influenza viruses.

IFITM3 is expressed at low levels in most cell types and is highly upregulated by interferons upon virus infection[5,6]. IFITM3 associates with cell membranes through a transmembrane domain and an *S*-palmitoylated amphipathic helix domain[6,7]. The amphipathic helix alters membrane curvature, lipid composition, and fluidity in a manner that disfavors virus-to-cell membrane fusion, endowing IFITM3 with the ability to block the release of virus genomes into the cytoplasm of cells[7–12]. As such, IFITM3 limits infection by a multitude of enveloped viruses in vitro, including influenza virus, Zika virus, and SARS-CoV-2[5,13–15]. In humans, there are two deleterious single nucleotide polymorphisms (SNPs) in the *IFITM3* gene that negatively impact its splicing or gene promoter efficacy[16–18]. These SNPs are unexpectedly common in the human population, with 20% of Chinese individuals being homozygous for the splicing SNP (rs12252-C), and 4% of Europeans being homozygous for the promoter SNP (rs34481144-A)[18]. Human IFITM3 deficiencies are associated with heightened severity of

[1]Department of Microbial Infection and Immunity, The Ohio State University College of Medicine, Columbus, OH, USA. [2]Viruses and Emerging Pathogens Program, Infectious Diseases Institute, The Ohio State University, Columbus, OH, USA. [3]Department of Infectious Diseases, St. Jude Children's Research Hospital, Memphis, TN, USA. [4]Department of Veterinary Preventive Medicine, The Ohio State University, Columbus, OH, USA. ✉e-mail: Jacob.Yount@osumc.edu

influenza, and data are emerging that IFITM3 deficiencies are a risk factor for severe COVID-19[16–20]. Indeed, we and other groups have observed that *Ifitm3*[−/−] mice experience more severe influenza virus and SARS-CoV-2 infections, mimicking human IFITM3 defects[15,16,21–23]. Interestingly, IFITM3 deficiencies in humans may have the most profound effects in emergent virus infections, e.g., pandemic 2009 H1N1 influenza virus[16,17,24], avian H7N9 influenza virus[19], HIV[25], and SARS-CoV-2[16–20]. Some studies examining seasonal influenza have not observed major impacts of the *IFITM3* SNPs on infection severity[26–28], suggesting that pre-existing adaptive immunity may compensate for diminished IFITM3 activity in some circumstances.

We hypothesized that IFITM3 deficiency removes a major block to initial zoonotic influenza virus infections and to subsequent viral replication, thus promoting virus evolution and the emergence of species-specific adaptive mutations. Herein, we demonstrate that IFITM3 deficiency not only lowers the infectious dose threshold for influenza viruses in mice and human cells but also promotes influenza virus adaptation to a new species. These functions of IFITM3 could not be inferred from its known roles, as virus passaging in the absence of interferon signaling, which also enhances virus replication, resulted in virus attenuation. Our work indicates that human IFITM3 deficiencies are a unique "Achilles heel" for the emergence of new pandemic viruses and should be considered in pandemic prevention efforts.

## Results

### IFITM3 deficiency lowers the minimum infectious dose of avian influenza viruses in mice

Intrinsic immunity is presumed to be among the factors responsible for preventing infection when virus inoculums are below a minimum infectious dose threshold. However, this fundamental concept lacks concrete experimental evidence. IFITM3 limits the severity of influenza virus infections in humans and mice[16,17,21–23], but whether it influences the minimum viral dose required for productive infection in vivo has not been tested. We infected WT and *Ifitm3*[−/−] mice with doses of 1, 10, or 50 TCID50 of H5N1 avian influenza virus and measured lung viral loads at day 3 post-infection. Infection with 1 TCID50 of H5N1 influenza virus resulted in significant viral replication in the lungs of *Ifitm3*[−/−] mice, approaching titers of $10^4$ TCID50/mL, while live virus was not detected in the lungs of WT mice (Fig. 1a). Doses of 10 and 50 TCID50 caused detectable viral loads in both WT and KO mice, with KO mice showing significantly higher viral titers at both doses (Fig. 1a). Examination of lung IL-6 and IFNβ as measures of inflammation induced by viral replication substantiated these titer results. Both lung IL-6 and IFNβ were nearly undetectable in WT mice infected with 1 TCID50 but were significantly above baseline in all other samples, including *Ifitm3*[−/−] mice infected with only 1 TCID50 (Fig. 1b, c). We next infected WT and KO mice with a second avian virus using doses of 1 and 10 TCID50. We again observed that 1 TCID50 of the H7N3 virus was sufficient to infect *Ifitm3*[−/−] mice as indicated by robust replication and induction of IL-6 and IFNβ, whereas the same viral dose did not productively infect WT mice (Fig. 1d–f). A dosage of 10 TCID50 resulted in productive virus replication in both WT and *Ifitm3*[−/−] mice, though viral titers and cytokine induction were each higher in the KO animals (Fig. 1d–f).

### IFITM3 deficiency lowers the minimum infectious dose of avian influenza viruses in human cells

To extend our conclusions to a human system, we tested a broad panel of influenza viruses isolated from animals in a relevant human cell type, A549 lung epithelial cells. For our study, we compared infection rates of 11 avian, 3 swine, and 2 human influenza viruses representing

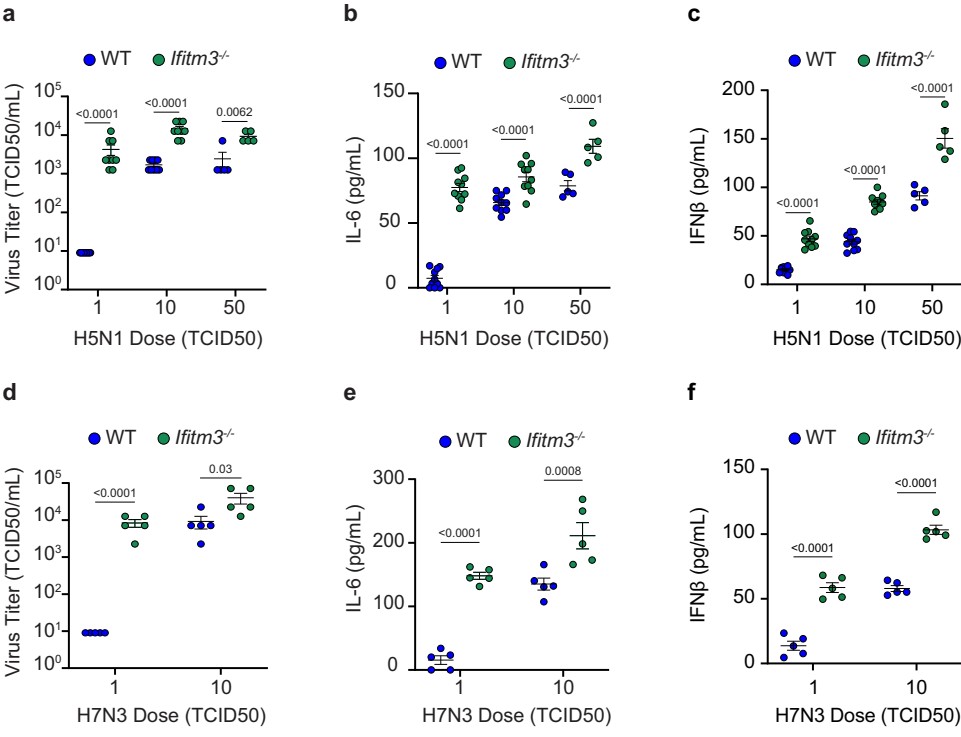

**Fig. 1 | IFITM3 deficiency lowers the minimum infectious dose threshold for avian influenza viruses in vivo.** WT and *Ifitm3*[−/−] mice were intranasally infected with (**a**–**c**) 1, 10, or 50 TCID50 of H5N1 avian influenza strain (2 independent experiments for doses 1 and 10 (*n* = 10 mice) and 1 experiment for dose of 50 (*n* = 5 mice)) or with (**d**–**f**) 1 or 10 TCID50 of H7N3 avian influenza strain (*n* = 5 mice). **a**, **d** Viral titers from lung homogenates at day 3 post infection. **b**, **e** ELISA quantification of IL-6 levels in lung homogenates at day 3 post infection. **c**, **f** ELISA quantification of IFNβ levels in lung homogenates at day 3 post infection. All error bars represent SEM. Comparisons were analyzed by one-way ANOVA followed by Tukey's multiple comparisons test. Only comparisons between WT and *Ifitm3*[−/−] mice for each dose are shown. **a**–**f** Each data point represents an individual mouse. The numbers shown above the graph represent exact *p*-values. Source data are provided as a Source Data file.

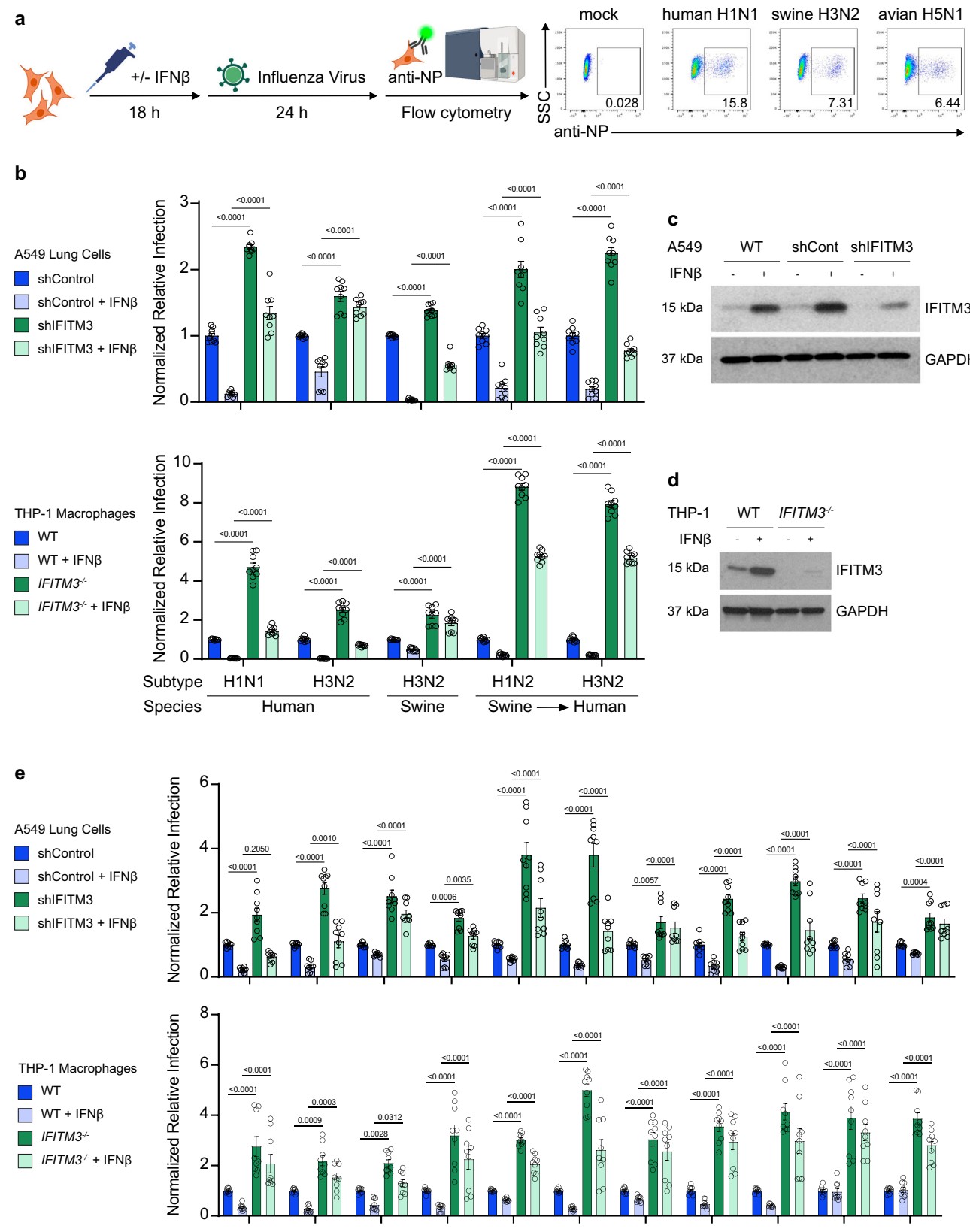

distinct and diverse hemagglutinin protein (HA) subtypes (strain details in Supplementary Table 1 and HA-based phylogenetic tree in Supplementary Fig. 1). Notably, 2 of the swine viruses are known to have infected humans making them confirmed zoonotic viruses[29,30]. We first examined single-cycle infections of these viruses in the

absence of trypsin in control A549 cells versus those in which IFITM3 was depleted by shRNA knockdown (Fig. 2a). The gating strategy for determining infected A549 cells is included in Supplementary Fig. 2a. The IFITM3-knockdown cells were universally infected at a higher rate (Fig. 2b, c, e and Supplementary Fig. 3), demonstrating that each of

**Fig. 2 | IFITM3 limits animal-origin influenza virus infection of human cells.**
**a** Schematic of in vitro infection with animal-origin influenza viruses and representative example flow cytometry dot plots from infected A549 human lung cells.
**b, e** The indicated A549 cells or THP-1 differentiated macrophages were treated +/-IFNβ for 18 h followed by infection with the indicated viruses (MOI 1) for 24 h.
Percent infection was determined by flow cytometry and normalized to respective control cells without IFNβ pre-treatment. Error bars represent SEM. Only statistical comparisons between shControl versus shIFITM3 and WT versus *IFITM3*[−/−] are

shown, determined by one-way ANOVA followed by Tukey's multiple comparisons test. Data are representative of 3 independent experiments, each performed in triplicate ($n = 9$). **c, d** Western blots of cell lysates at 18 h +/− IFNβ treatment. Note that commercial IFITM3 antibodies weakly detect IFITM2 in addition to IFITM3. The numbers shown above the graph represent exact p values. Source data are provided as a Source Data file. **a** Created in BioRender. Denz, P. (2024) BioRender.com/z17v580. Gating strategies for flow cytometry are depicted in Supplementary Fig. 2a. A549 cells and Supplementary Fig. 2b. THP-1 cells.

these diverse viruses was restricted by IFITM3 in human cells. Furthermore, IFITM3-knockdown cells maintained higher infection susceptibility than control cells after treatment with type I interferon (Fig. 2b, c, e and Supplementary Fig. 3), demonstrating that IFITM3 plays a critical and non-redundant role in the human antiviral interferon response against diverse influenza viruses. Since macrophages are also a relevant target of influenza virus[31], we examined infection of differentiated human THP-1 cells. Once again, infection by the human-, avian-, and swine-origin viruses was significantly potentiated in the human macrophages lacking IFITM3 with or without interferon treatment (Fig. 2b, d, e; Supplementary Fig. 2b and Supplementary Fig. 4). Increased infection was also observed for IFITM3 KO HAP1 fibroblasts (Supplementary Fig. 2c and Supplementary Fig. 5) and for IFITM1/2/3 KO HeLa cells (Supplementary Fig. 2d, Supplementary Fig. 6). Of note, we determined sialic acid receptor binding preferences for viruses we tested and found that the human- and swine-origin viruses showed a preference for a-2,6-linked sialic acid while many of the avian viruses showed a preference for a-2,3-linked sialic acid (Supplementary Table 1). Despite these differences in receptor binding preference, all the viruses we examined were able to infect every cultured human cell type that we tested (Fig. 2b, e and Supplementary Figs. 2–6). Overall, these data demonstrate that IFITM3 deficiency broadly increases infection of human cells with influenza viruses of animal origin.

To further examine the impact of IFITM3 on minimum infectious dosing, we infected A549 lung cells with or without IFITM3 knockdown using H5N1 or H7N3 avian influenza viruses at a range of doses from MOI 0.001 to 10. We observed that infected cells could be detected at lower viral doses for both H5N1 (Fig. 3a) and H7N3 (Fig. 3b) when IFITM3 was deficient. To further examine the effects of IFITM3, we infected both control and IFITM3-knockdown A549 cells with a selection of viruses, including 2 human-origin, 3 swine-origin (2 with known zoonotic transmission[29,30]), and 2 avian-origin viruses at an MOI of 1. The cells were incubated with TPCK-trypsin to allow for multiple rounds of infection, and at 48 h post infection, we collected the supernatants to determine viral titers by TCID50 assay. For all viruses, we measured significantly higher titers produced by IFITM3-knockdown versus control cells (Fig. 3c). Along with data from Fig. 1, these data establish that IFITM3 prevents animal-origin influenza virus infection in vitro and in vivo when the virus dose is below a minimum threshold and that influenza virus replicates to higher titers in the absence of IFITM3. These data overall demonstrate that IFITM3 increases the minimum infectious dose necessary to achieve a productive influenza virus infection.

## IFITM3 deficiency facilitates the adaptation of influenza viruses to a new host species in vivo

Several factors have been proposed to favor inter-species adaptation of a virus to a new host, including levels of virus replication and infectious doses. Given that IFITM3 deficiency impacts both of these features, we tested whether its absence allows influenza virus to adapt more readily in vivo. With biosafety in mind, we did not attempt to adapt avian influenza virus to a mammalian host since such adaptations could increase virus replication across mammalian species, including humans, and could thus be considered dual-use research of concern[32]. Rather, we sought to adapt viruses of human origin to mice,

which has been performed safely for nearly a century in the influenza virus research field[32]. Initially, we performed lung-to-lung passaging of influenza virus strain A/Victoria/361/2011 (H3N2) 10 times through WT or *Ifitm3*[−/−] mice using intranasal infection and allowing 3 days for the virus to replicate between successive passages (Fig. 4a). We chose this strain because preliminary experiments indicated that it infects mice, but replicates to lower levels than commonly used mouse-adapted strains, thus providing a virus with significant potential for adaptation to mice.

We propagated passages 1, 5, and 10 from both WT and *Ifitm3*[−/−] mice, as well as the parental virus stock (passage 0), in MDCK cells. We then used these expanded stocks to challenge WT mice with equal virus doses. 1000 TCID50 of passages 1, 5, and 10 from WT and KO mice, along with parental virus (passage 0), were used to infect WT mice for 7 days, at which point mice were sacrificed to measure viral titers and inflammatory cytokines (Fig. 4b). To reiterate, passaged viruses, whether generated in WT or KO mice, were tested for adaptation in WT animals. We found that, as expected, the ability to replicate in mouse lungs was unchanged for the earliest passage of virus from WT or KO mice as compared to the parental virus (Fig. 4c). However, passage 5 from KO mice exhibited an upward trend in viral titers and KO passage 10 showed a statistically significant, >1 log average increase in lung viral titers (Fig. 4c). In contrast, virus passages 5 and 10 from WT mice did not show significantly enhanced replication capacity compared to the parental virus (Fig. 4c). As an independent metric for corroborating virus replication, we measured levels of the inflammatory cytokines IL-6 and IFNβ in lungs during infection, which mirrored viral loads, with IFITM3 KO passage 10 inducing a statistically significant increase in both IL-6 (Fig. 4d) and IFNβ (Fig. 4e) compared to infection with parent virus. Since virus adaptation is a stochastic process, we independently repeated our H3N2 virus passaging and testing a second time and obtained similar results indicating enhanced adaptation occurring in IFITM3 KOs (Fig. 4f–h). Interestingly, the animals infected with both of the H3N2 virus passage series did not lose weight despite increased viral titers seen for the IFITM3 KO-passaged virus stocks, suggesting only partial adaptation. We also tested the passaged virus stocks for replication fitness in A549 human lung cells versus LET1 mouse lung cells. We found that the IFITM3 KO-passaged viruses showed increased replication in the murine LET1 cells compared to the parent virus, but replication in human A549 cells was not changed (Supplementary Fig. 7). These results are in agreement with expectations that mouse adaptation would not provide a gain of function for infection and replication in human cells. Overall, our data from both passage series demonstrate that influenza virus gained enhanced replication capacity and induction of inflammation when passed through IFITM3 deficient versus WT hosts.

We next tested whether IFITM3 deficiency is unique in its ability to facilitate virus adaptation or whether passaging in another immune-compromised system that allows enhanced virus replication would also promote interspecies adaptation. The interferon system is particularly important in this regard as interferons widely inhibit virus replication[33]. Further, it has become increasingly apparent throughout the recent COVID-19 pandemic that genetic and antibody-mediated interferon deficiencies are significantly more common in the human population than previously appreciated[34,35]. We thus repeated our

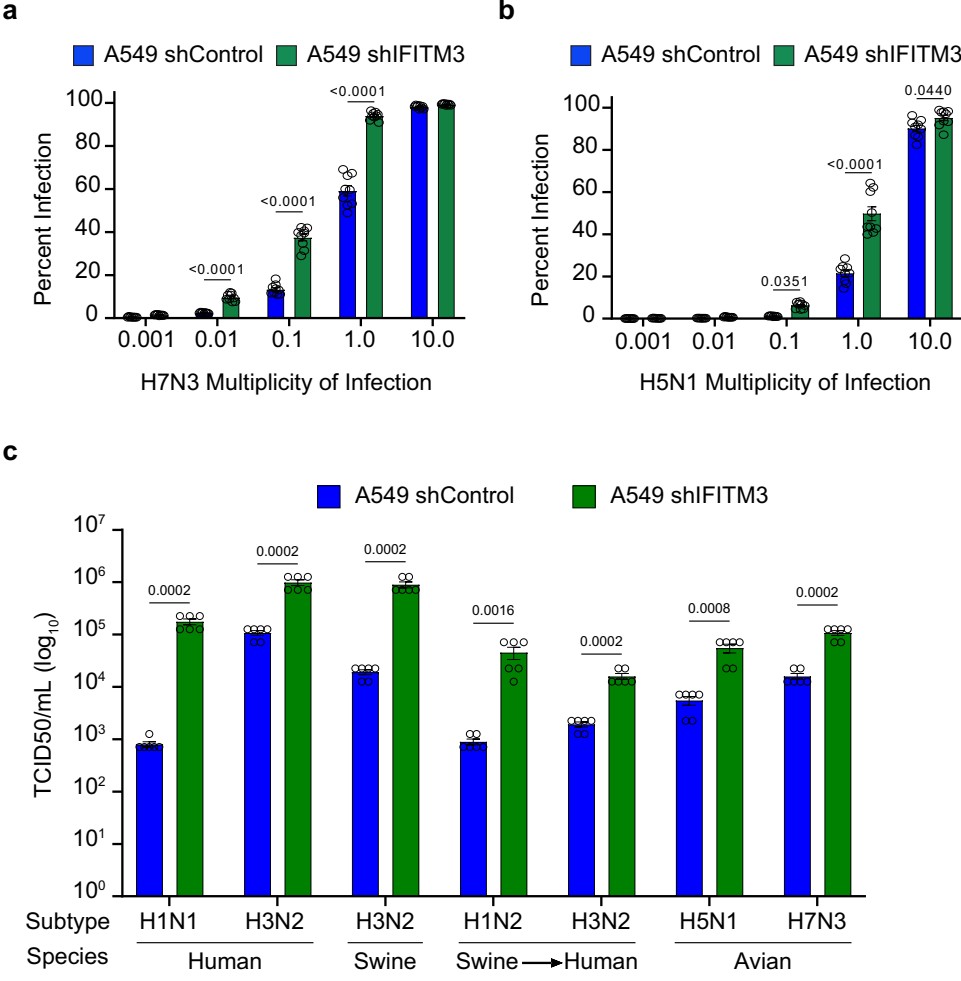

**Fig. 3 | IFITM3 deficiency lowers the minimum infectious dose threshold for avian influenza viruses in vivo. a, b** The indicated A549 cells were infected with the indicated avian-origin influenza viruses at a range of 0.001 to 10 multiplicity of infection (MOI) for 24 h. Percent infection was determined by flow cytometry. Data are representative of 3 independent experiments, each performed in triplicate ($n = 9$). Only comparisons between control and knockdown cells are shown at each dose. **c** A549 cells were infected with the indicated influenza viruses at an MOI of 1 and incubated in media containing TPCK-Trypsin for 48 h to allow for multi-cycle replication. The supernatants were collected to determine viral titer. Data are representative of 2 independent experiments, each performed in triplicate ($n = 6$). Only comparisons between control and knockdown cells are shown for each virus. All error bars represent SEM. The numbers shown above the graph represent exact $p$-values. Comparisons were analyzed by one-way ANOVA followed by Tukey's multiple comparisons test. Source data are provided as a Source Data file.

passaging and challenge regimen with influenza virus A/Victoria/361/2011 (H3N2), comparing passaging through WT versus *Stat1*$^{-/-}$ mice, which lack signaling downstream of all interferon receptors (Supplementary Fig. 8a, b). Consistent with our previous results, we did not observe significant adaptation of the virus passed through WT mice (Supplementary Fig. 8c). In contrast, passage 5 from the STAT1 deficient mice exhibited a downward trend in viral replication and *Stat1*$^{-/-}$ passage 10 showed a statistically significant reduction in the ability to replicate in WT mice (Supplementary Fig. 8c). The titer data were supported by measurement of the inflammatory cytokine IL-6 in that STAT1 KO passage 10 showed a significant decrease in IL-6 levels compared to other groups (Supplementary Fig. 8d). Remarkably, IL-6 levels were uncoupled from IFNβ levels (Supplementary Fig. 8e), which were highest for the STAT1 KO-passaged virus, suggesting that this stock induces increased IFNβ, which may account for decreased virus replication. Additionally, when testing the passaged virus's ability to replicate in A549 versus LET1 cells, we saw that STAT1 KO-passaged viruses showed reduced replication in LET1 cells compared to the parent virus, with two out of six samples showing replication below detection limits, while replication in A549 cells was unchanged (Supplementary Fig. 8f, g) Our findings are consistent with the principle

that, in the absence of selective pressures from the interferon system, viral interferon antagonism mechanisms become less active, resulting in virus attenuation when infecting WT systems. Importantly, these results underscore how IFITM3 may be a unique vulnerability in innate immune response pathways that, when impaired, can promote viral adaptation to new hosts rather than viral attenuation.

To further investigate the generality of our findings on IFITM3, we repeated the passaging regimen in WT and *Ifitm3*$^{-/-}$ mice using pandemic influenza virus strain A/California/04/2009 (H1N1), followed by adaptation testing in WT mice. Both WT passage 5 and 10 virus stocks were modestly enhanced in replicative capacity (Fig. 5a). We saw that both WT and KO passage 10 virus stocks induced increased weight loss in infected mice compared to the parent H1N1, with the IFITM3 KO passage 10 mice inducing the most dramatic loss of body weight (Fig. 5b). WT passage 5 and 10 virus stocks exhibited no significant changes in their IL-6 and IFNβ induction capacity (Fig. 5c, d). Consistent with our previous results, virus passaged 10 times in *Ifitm3*$^{-/-}$ mice replicated to higher titers compared to the parental strain and to the WT mouse passaged viruses (Fig. 5a). In addition to the highest viral titer, KO passage 10 also induced robust levels of IL-6 and IFNβ in infected lungs (Fig. 5c, d). In a second series of passages with the H1N1

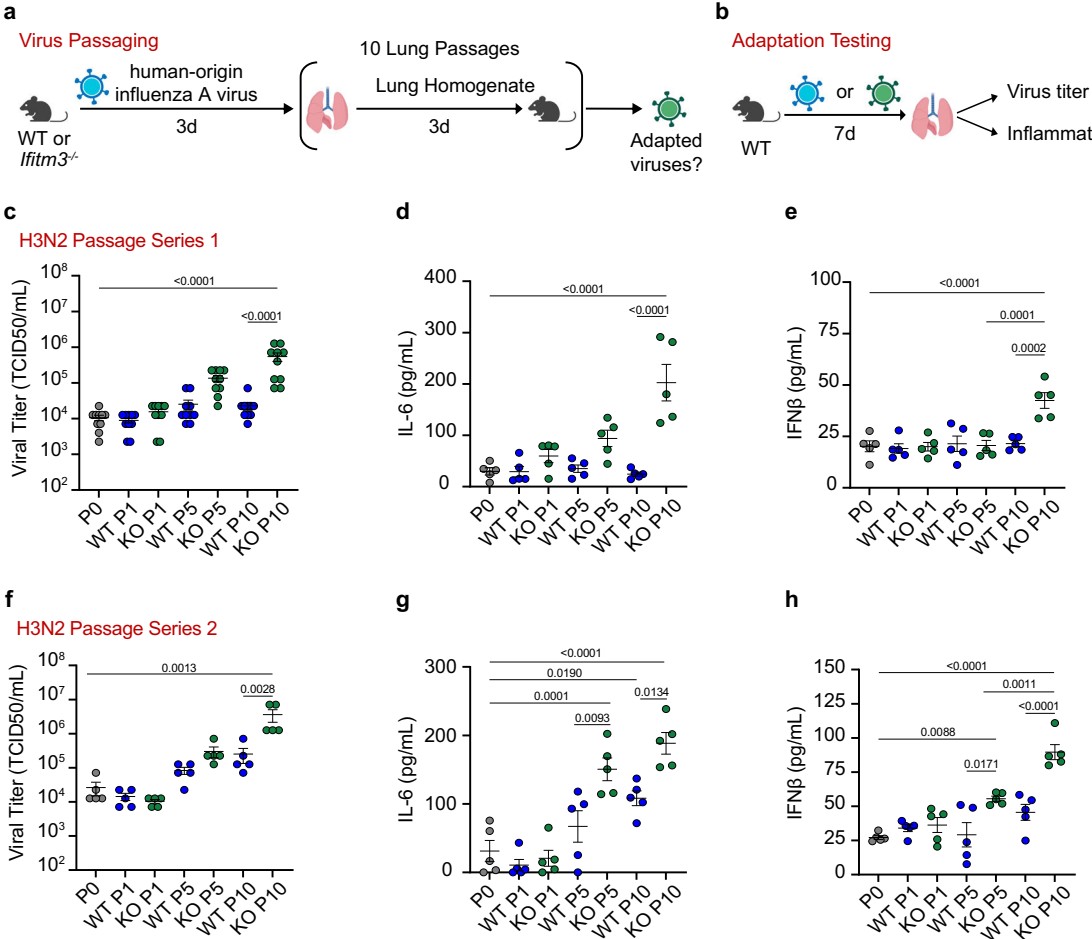

**Fig. 4 | Human-origin H3N2 influenza virus adapts to a new species more readily in the absence of IFITM3. a** Schematic of mouse passaging experiments. Initial intranasal infections were performed with 1000 TCID50 of parental viruses. **b** Schematic of WT mouse challenge with parental A/Victoria/361/2011 (H3N2) or passaged viruses. **c–h** Groups of WT mice ($n = 5$ per group) were challenged with 1000 TCID$_{50}$ of A/Victoria/361/2011 (H3N2) virus passaged 1, 5, or 10 times through WT or *Ifitm3*$^{-/-}$ mice and compared to the parent virus (passage 0). **c, f** Viral titers

from lung homogenates collected at day 7 (**c** represents 2 independent infections). **d, g** ELISA quantification of IL-6. **e, h** ELISA quantification of IFNβ. **c–h** Error bars represent SEM and comparisons were analyzed by one-way ANOVA followed by Tukey's multiple comparisons test. Each dot represents an individual mouse. The numbers above the graph represent exact *p*-values. Source data are provided as a Source Data file. **a, b** Created in BioRender. Denz, P. (2024) BioRender.com/h97v852.

---

virus, the most robust adaptation in terms of viral replication, weight loss, and inflammatory cytokine induction was again seen for KO passage 10 (Fig. 5e–h). Notably, this adapted viral stock became highly virulent, with all infected mice meeting humane endpoint criteria of >30% weight loss by day 6 post infection (Fig. 5f). Similar to our results with the H3N2 passages, infecting human A549 versus murine LET1 lung cells with the H1N1 passages showed increased replication and IFNβ induction in LET1 cells for the IFITM3 KO-passaged virus whereas we did not observe increased replication in the human cells (Supplementary Fig. 9), again confirming that mouse adaptation of human influenza viruses does not result in a gain of function in terms of increasing virus replication in human cells.

To investigate the specific changes within the passaged viruses that may contribute to the observed mouse adaptation, we extracted viral RNA and sequenced the genomes of passages 5 and 10 from all H1N1 and H3N2 series with comparison to parental viruses. Despite enhanced replication of the H3N2 IFITM3 KO passage 10 viruses, we did not observe changes to the consensus sequence of those viruses (>50% prevalence). Instead, we observed the emergence of minor variants, particularly in polymerase segments, that may contribute to the observed phenotypes (Supplementary Data 1, 2). Notably, the second H3N2 passage series showed a trend toward adaptation for the

virus passaged 10 times through WT mice in terms of virus replication, though this was not to the same magnitude as the IFITM3 KO-passaged virus and was not statistically significant. This may also be due to the contribution of minor variants present within the quaispecies of that passaged virus (Supplementary Data 1 and 2). The H3N2 virus passaged through *Stat1*$^{-/-}$ mice also showed the emergence of minor variants in polymerase segments that are distinct from those present in the IFITM3 KO-passaged viruses (Supplementary Data 1 and 3). These results are consistent with the H3N2 stocks being in the early stages of adaptation as supported by a lack of weight loss when infecting with these viruses and titers that, while increased for IFITM3 KO-passaged viruses, did not reach the levels achieved by the adapted H1N1 stocks.

In contrast to the H3N2 sequences, the H1N1 passages showed clear changes within the consensus sequences in addition to minor variants (Supplementary Data 4 and 5). The trend toward enhanced replication of WT passages 5 and 10 in passage series 1 was associated with two amino acid changes in the viral NP (D101G, R102G) and a single change in PB2 (E158G) that is a known polymerase activity-increasing adaptive mutation in mice[36,37]. The H1N1 series 1 KO passages 5 and 10 possessed the same adaptive PB2 E158G mutation seen in WT passages 5 and 10. However, the heightened replication and increased induction of inflammatory cytokines by KO passage 10

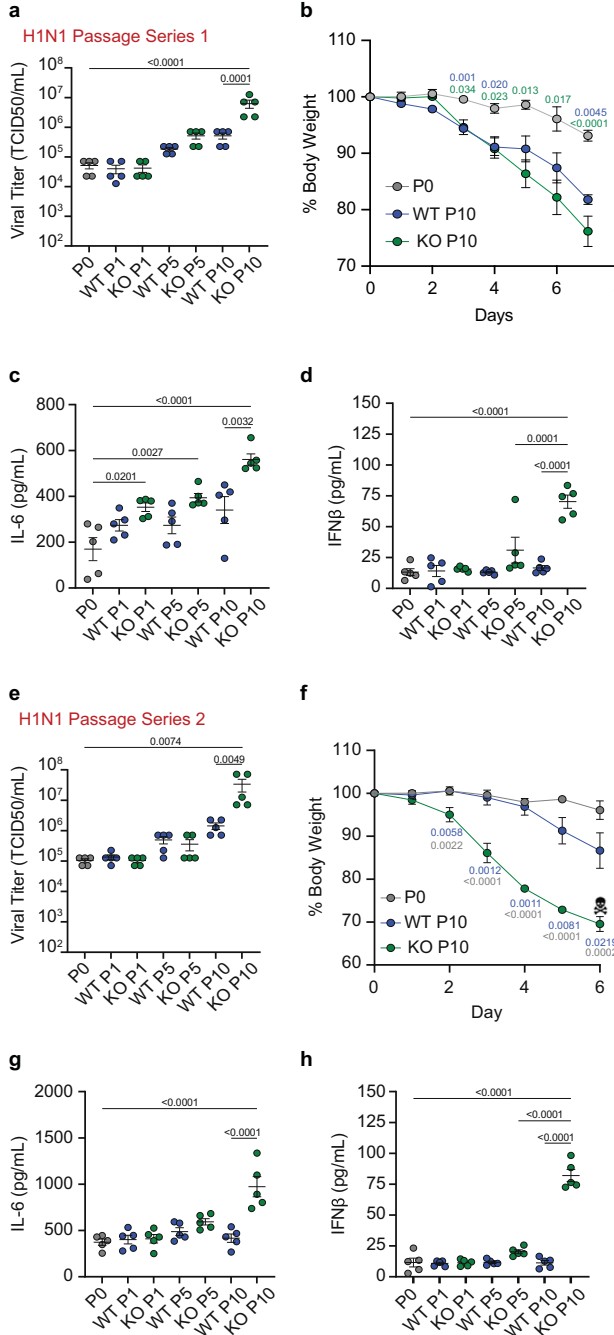

relative to KO passage 5 was associated with a single change in the PA polymerase subunit (N373H), representing a novel mutation not previously reported in mouse adaptation studies (Supplementary Data 4). The high virulence of H1N1 series 2 KO passage 10 was linked to a single consensus sequence change, specifically an E158A mutation in PB2 (Supplementary Data 5) that, similarly to the PB2 E158G mutation detected in our first series of H1N1 passages, is a known mouse-adaptive mutation[37]. Overall, our examinations of replication levels and weight loss, along with sequencing data, suggest that we've captured the two viruses (H3N2 and H1N1) at different stages of adaptation, though in each case, this process occurs more rapidly in IFITM3 KO mice. Together with data for the H3N2 virus, our H1N1 passaging experiments demonstrate that the adaptation of influenza viruses to a new host species is accelerated in the absence of IFITM3, fueled by mutations that readily emerge in the compromised host.

## Discussion

Three fundamentally important discoveries regarding IFITM3 emerged from our study. First, IFITM3 raises the minimum infectious dose of influenza virus required to achieve a detectable infection in human cells, as well as a productive infection in vivo. Second, IFITM3 serves as a critical barrier to pandemic virus emergence by limiting the adaptation of influenza viruses to new host species, whereas global defects in interferon responses lead to virus attenuation. Third, IFITM3, whether basally expressed or induced by interferon, broadly restricts infection of human cells by potentially zoonotic influenza viruses, and this function was independent of viral receptor binding preferences. Each of these findings supports the premise that human IFITM3 deficiencies represent a critical vulnerability for the entry and adaptation of animal-origin influenza viruses in the population.

While we did not perform virus transmission experiments, our work indirectly addresses the transmission of influenza viruses. Mice are unable to transmit influenza virus to one another, even when in close contact[32]. We have also observed in unpublished work that IFITM3 KO mice, perhaps the most susceptible mouse model of influenza, do not transmit mouse-adapted influenza virus or mouse-adapted SARS-CoV-2 to non-infected, cohoused KO animals. Nonetheless, we found that the minimal in vivo infectious dose of avian influenza virus, a critical aspect of the transmission process, was lowered in the absence of IFITM3. Specifically, $Ifitm3^{-/-}$ animals showed significant replication of the virus in their lungs after infection with only 1 TCID50 of H5N1 or H7N3 virus, while titers were undetectable in WT lungs when infected at the same dose. These results provide experimental evidence for the long-presumed concept that minimum infectious doses of the virus are set by levels of resistance from the innate immune system. Specifically, we show that IFITM3 underlies sterilizing immunity that prevents infection at low influenza virus doses that are below a minimum threshold.

We designed our virus passaging experiments to model transmission of a zoonotic virus through a family or community in which deleterious *IFITM3* SNPs are prevalent. One caveat to this approach is that, while our engineered mice are fully devoid of IFITM3 production, deleterious SNPs in human *IFITM3* substantially reduce IFITM3 levels or activity, but do not ablate the protein[16,17]. In this regard, lung cells partially deficient in IFITM3 (IFITM3-knockdown A549 cells in Fig. 2) were indeed significantly more susceptible to diverse influenza virus strains. While infections in humans with IFITM3 deficiencies may represent more nuanced scenarios, experimentation with complete KO mice allowed us to unambiguously identify important

**Fig. 5 | Human-origin H1N1 influenza virus adapts to a new species more readily in the absence of IFITM3.** Influenza virus A/California/04/2009 (H1N1) was passaged through mice as described in Fig. 4a. Groups of WT mice (*n* = 5 per group) were challenged with 1000 TCID₅₀ of A/California/04/2009 (H1N1) virus passaged 1, 5, or 10 times through WT or *Ifitm3*⁻/⁻ mice and compared to the parent virus (passage 0). **a, e** Viral titers from lung homogenates collected at day 7 (**a**) or day 6 (**e**) post infection. Error bars represent SEM, comparisons were analyzed by one-way ANOVA followed by Tukey's multiple comparisons test. **b** Weight loss for the H1N1 series 1 challenge. Error bars represent SEM, comparisons were made using the Mann-Whitney test. ELISA quantification of IL-6 (**c, g**) and IFNβ (**d, h**) levels in lung homogenates of WT and IFITM3 KO mice at day 7 (**c, d**) or day 6 (**g, h**) post infection. Error bars represent SEM and comparisons were analyzed by one-way ANOVA followed by Tukey's multiple comparisons test. **f** Weight loss for the H1N1 series 2 challenge. Skull and crossbones indicate humane euthanasia of all animals infected with KO passage 10. Error bars represent SEM, comparisons were made using the Mann-Whitney test. (**a, c, d, e, g, h**) Each dot represents an individual mouse. **b, f** dots represent averages of individual mice (*n* = 5 per group). All numbers above the graphs represent exact *p*-values. Source data are provided as a Source Data file.

and relevant functionality of IFITM3 in limiting interspecies infection and adaptation of viruses to a new host species in vivo, key findings that are likely germane to preventing pandemics in humans.

It is also interesting to note that distinct *IFITM3* SNPs were independently selected in both Chinese and European populations. While the high prevalence of *IFITM3* SNPs in these groups is puzzling, given their association with increased viral disease severity, a negative role for IFITM3 in placental development may explain their selection and maintenance in these populations. Indeed, IFITM3 prevents the endogenous retrovirus fusogen-mediated cell-to-cell fusion of placental trophoblasts required for proper placental architecture and function[38,39]. In this way, IFITM3 mediates a significant portion of the fetal toxicity of interferons produced during pregnancy-associated infections[38,40]. Increased fetal survival during certain infections in the absence of IFITM3 thus represents the prevailing theory for how IFITM3 deficiencies may provide a selective benefit despite increasing susceptibility to emergent virus infections later in life.

RNA viruses generally adapt to new host species by producing mutations arising from error-prone polymerases[41]. Beneficial mutations that allow the virus to replicate more efficiently are enriched by outcompeting the parental virus and other mutants. Indeed, influenza virus and several other viruses replicate to higher titers in *Ifitm3*[−/−] versus WT mice[15,16,21–23,42–44], which represents a plausible mechanism for enhanced adaptation in these animals[41]. Similarly, our results indicate that IFITM3 deficiency also relaxes the infection bottleneck present at initial inoculation. In our current study, mutations in the 2009 pandemic H1N1 virus passages that drive adaptation in mice were primarily observed in the viral polymerase subunits. Indeed, PB2 mutations at the E158 position are consistently reported in mouse-adapted versions of this virus[36,37]. Moreover, the relative impact of E158 mutations on polymerase activity is in accord with viral replication and virulence levels observed in our in vivo experiments. We also identified a novel N373H mutation in PA from one H1N1 passage series, which emerged subsequent to the E158G mutation. The dual mutations were associated with enhanced replication in vivo compared to E158G alone. The mechanistic impact of the PA N373H mutation remains to be determined and may provide future insights into influenza virus replication and adaptation. Interestingly, we do not see an overall increase in variants in our IFITM3 KO-passaged virus stocks. Rather, in all four of our IFITM3 KO passaging experiments, the stock generated from the 10th KO passage contains the fewest number of minor variants as compared to parental virus and WT passages. This suggests that IFITM3 deficiency may promote viral adaptation by allowing adaptive variants to more efficiently outcompete other viruses in the quasispecies, perhaps by allowing infection of more cells per animal as our in vitro experiments would suggest.

The prevalence of genetic and immunological mechanisms that impair interferon induction or signaling in humans garnered increased attention during the COVID-19 pandemic since 20% of all severe COVID-19 cases could be attributed to interferon defects[34,35]. Given that IFITM3 expression is induced by interferon, such defects likely diminish its induction, as well as that of other antiviral proteins, which could allow increased replication of viruses and enhanced adaptation to humans. However, passaging through interferon deficient systems often results in attenuated viruses that have lost interferon antagonism due to a lack of selective pressure for their maintenance[45,46]. Indeed, our passaging of influenza virus in *Stat1*[−/−] animals resulted in virus attenuation, despite influenza virus replicating

to high titers in interferon deficient systems[47]. Additional studies on viral evolution in human populations and engineered animals will be required to address how broad interferon defects mechanistically influence the adaptive evolution of influenza virus and other pathogens.

While IFITM3 inhibited infection by all the influenza viruses in all cell types in our experiments, we observed that the degree of inhibition varied to some extent depending on the individual viruses. Published work has suggested that differences of distinct influenza virus isolates in susceptibility to IFITM3 may be due to differences in viral fusion pH optimums leading to fusion preferences in early or late endosomes where IFITM3 levels differ[48]. Importantly, influenza viruses capable of fully evading IFITM3 restriction have not been identified to date. Likewise, unlike the well-characterized antagonism of interferon by the influenza virus NS1 protein[47], influenza virus antagonism of IFITM3 has not been reported. Thus, in contrast to interferon, IFITM3 is unlikely to contribute similar selective pressures on influenza virus, allowing the virus to efficiently adapt in its absence. IFITM3 deficiency may, therefore, represent a particularly unique vulnerability for the entry and adaptation of viruses in a new host species. Broad testing of IFITM3 status in the human population could bolster pandemic prevention efforts by allowing vulnerable individuals to receive greater vaccine coverage or exercise enhanced precautions when encountering animal reservoirs of infection.

## Methods
Our research complies with all relevant ethical regulations and was approved by the Institutional Biosafety Committee (IBC) and Institutional Animal Care and Use Committee (IACUC) of The Ohio State University under protocol number 2016A00000051-R2.

### Cell culture
Control and IFITM3 knockdown A549 cells were generated by lentiviral shRNA-mediated targeting using previously described constructs and methods[49]. THP-1 IFITM3 knockout cells were generated via CRISPR-Cas9 targeting (provided by Dr. Anasuya Sarkar of the Ohio State University). HeLa IFITM1/2/3 knockout and HAP1 IFITM3 knockout were purchased from ATCC (CRL-3452) and Horizon Discovery Biosciences (HZGHC004186c010), respectively. MDCK cells were obtained from BEI Resources (NR-2628). LET1 cells were also obtained from BEI resources (NR-42941). Cells were maintained in DMEM medium (Fisher Scientific) supplemented with 10% EquaFETAL bovine serum (Atlas Biologicals) except for the THP-1 cells, which were maintained in RPMI 1640 medium (Fisher Scientific) supplemented with 10% EquaFETAL bovine serum (Atlas Biologicals). All cells were cultured in a humidified incubator at 37 °C with 5% CO2.

### Virus stocks and in vitro infection
Influenza viruses A/Puerto Rico/8/34 (H1N1) (PR8, provided by Dr. Thomas Moran of the Icahn School of Medicine at Mt. Sinai), A/Victoria/361/2011 (H3N2) (BEI Resources), and A/California/4/2009 (H1N1) (BEI Resources) were propagated in 10-day embryonated chicken eggs (Charles River Laboratories) and titered on MDCK cells. The animal-origin viruses were collected by Andrew Bowman of the Ohio State University and were also propagated in embryonated chicken eggs (Charles River Laboratories) and titered on MDCK cells. Infection of cell lines with the panel of animal-origin viruses (Fig. 2) was done at an MOI of 1 for 24 h in infection media without trypsin and represents a single-cycle infection. Cells that received Interferon-β pre-treatment were stimulated for 18 h with 100 units of recombinant human Interferon-β protein (Millipore, IF014) prior to infection. All titration of virus stock was done using a TCID50 assay on MDCK cells, as previously described. Cells incubated with TPCK-Trypsin (Worthington Biochemical Corporation,

LS02123) were treated with 2 µg/mL for 48 h to allow for multi-cycle replication (Fig. 3 and Supplementary Figs 6-8).

### Determination of virus receptor binding preferences

50 µL of horse (1%) or turkey (0.5%) RBCs (Innovative Research Inc, IHSRBC100P30M, and ITKRBC5P50ML, respectively) were added to 50 µL of 2-fold dilutions of virus stocks in PBS and incubated for 1 h at room temperature to determine the hemagglutination titer for each virus. The turkey RBCs were either neuraminidase-treated to remove the α−2,3 sialyl linkages from glycoproteins and glycolipids or mock-treated. For neuraminidase treatment, 100 µL of a 5% Turkey RBC suspension prepared in PBS was treated with 15 units of α−2,3 neuraminidase S (New England Biolabs, P0743) for 1 h at 37 °C. The treated TRBCs were then diluted with PBS to a working concentration of 0.5% for the HA assay.

### Mouse studies

All mice used in this study were of the C57BL/6 J background. *Ifitm3*$^{-/-}$ mice with a 53 base pair deletion in exon 1 of the *Ifitm3* gene were described previously[23]. WT mice for comparison to *Ifitm3*$^{-/-}$ mice were obtained from Charles River Laboratories. *Stat1*$^{-/-}$ mice (Strain #: 012606) and complementary WT C57BL/6J mice (strain #007914) were obtained from The Jackson Laboratory. Mice were housed in the Ohio State University's Biomedical Research Tower vivarium, which is maintained at 68−76 degrees F, with a 12:12 light−dark cycle and humidity between 30−70%. Autoclaved individually ventilated cages (Allentown) were used for housing. Mice were fed irradiated natural ingredient chow ad libitum (Evnigo Teklad Diet 7912). Reverse osmosis purified water was provided through an automated rack water system. Cages included ¼ inch of corn cob bedding (Bed-o-Cobs, The Andersons) with cotton square nesting material. Male and female mice between 6 and 10 weeks of age were used in our experiments. All mice in each individual experiment were age-matched. All mouse infections were performed intranasally under anesthesia with isoflurane, delivering 25 µL of inoculum per nare. For virus passaging, mice were initially infected with doses of 1000 TCID50, and mouse organs were collected and homogenized in 500 µL of sterile saline. Propagation and titering of passaged viruses, as well as parental control viruses, was done on MDCK cells. WT mice were infected with equal doses (1000 TCID50) for testing of virus adaptation. Innoculums were prepared via dilution of viral stock in sterile PBS. For determining lung titers and cytokine levels in virus adaptation studies, lungs were collected and homogenized in 1 mL of PBS, flash-frozen, and stored at −80 °C prior to titering on MDCK cells or analysis via ELISA. All procedures were approved by the OSU IACUC and were performed in accordance with guidelines for the ethical use of animals.

### Virus sequencing

Viral RNA was extracted from MDCK-grown stocks using the RNeasy Mini Kit (Qiagen), and cDNA was synthesized using the Superscript IV First-Strand Synthesis System (Invitrogen). The influenza A virus gene segments were amplified using modified universal primers in a multisegment PCR as previously described[50]. PCR products were purified using Agencourt AMPure XP beads according to the manufacturer's protocol (BeckmanCoulter). Libraries were prepared using the Nextera XT DNA Library Prep Kit (Illumina) according to the manufacturer's protocol and sequenced using a MiSeq Reagent Kit v2 (300 cycles) on a MiSeq System (Illumina). Sequencing reads were then quality trimmed and assembled using CLC Genomics Workbench (version 22.0.1).

### Phylogenetic analysis

The HA gene segments of the individual viruses used in this study were downloaded from Genbank and used for analysis. Alignments and tree estimates for whole HA genomic segments were performed using Molecular Evolutionary Genetics Analysis (MEGA) software v11.0.13.

The sequences were aligned using the MUSCLE alignment. This was used to generate a neighbor-joining tree.

### Western blotting

For detection of protein expression in cell lines, samples were lysed in 1% SDS buffer (0.1 mM triethanolamine, 150 mM NaCl, 1% SDS (Sigma), pH 7.4) containing Roche Complete Protease Inhibitor Cocktail. The lysates were centrifuged at 16,000 × *g* for 10 min, and soluble protein supernatants were used for Western blot analysis. Equal amounts of protein (30 µg) were separated by SDS-PAGE and transferred onto membranes. Membranes were blocked with 10% non-fat milk in Phosphate-buffered saline with 0.1% Tween-20 (PBST) and probed with antibodies against IFITM1 (Cell Signaling Technology, #13126), IFITM2 (Cell Signaling Technology, #13530), IFITM3 (ProteinTech, 11714-1-AP), and GAPDH (Thermo Scientific, ZG003). Antibodies were used at dilutions of 1:1000 in PBST as determined by preliminary experiments optimizing the protocol for western blotting.

### ELISAs

ELISAs were performed on supernatant from mouse lung homogenates using the Mouse IL-6 Duoset ELISA kit from R&D Systems (catalog # DY406) or the Mouse IFNβ Duoset ELISA kit from R&D Systems (catalog # DY8234-05) according to manufacturer's instructions.

### Flow cytometry

Cells were fixed with 4%paraformaldehyde for 10 min at room temperature, permeabilized with 0.1% Triton X-100 in PBS, stained with an antibody against Influenza A Virus Nucleoprotein (Abcam, ab20343) conjugated with Goat anti-Mouse IgG (H + L) Highly Cross-Adsorbed Secondary Antibody, Alexa Fluor™ 647 (Thermo Scientific, Catalog # A-21236) resuspended in 2% FBS in PBS, and run on a BD FACS Canto II flow cytometer. Antibodies were used at dilutions of 1:1000 in 0.1% Triton X-100 in PBS as determined by preliminary experiments optimizing the protocol for flow cytometry. Results were analyzed using FlowJo Software (version 10.8.1).

### Reporting summary

Further information on research design is available in the Nature Portfolio Reporting Summary linked to this article.

## Data availability

All data are available in the main text or the supplementary materials. Source data are provided in this paper. The sequencing data generated in this study have been deposited as the consensus sequence in the GenBank database under accession codes PQ384645−PQ384820. Source data are provided in this paper.

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

## Acknowledgements

The authors thank Dr. Eugene Oltz (Ohio State University) for critical reading and editing of the manuscript. We also thank Dr. Ana Sarkar and Paul Consiglio (Ohio State University) for providing IFITM3 KO THP-1 cells. Research in the Yount Laboratory is funded by NIH grants AI130110, HL168501, HL157215, HL154001, and AI151230. Research in the Hemann laboratory is funded by NIH grant AI146141. A.D.K. and P.J.D. were funded by NIH training grant AI112542.

## Author contributions

P.J.D. organized and performed all experiments in the manuscript, analyzed data, and contributed to manuscript writing and editing of the manuscript. S.S., A.D.K., A.C.E., J.L.P., J.R., and S.S. assisted with mouse colony genotyping/maintenance, virus titering, and mouse experiments. E.A.H. and A.F. provided conceptual input and experimental design and edited the manuscript. R.J.W. and A.R. performed 2009 H1N1 and 2011 H3N2 virus sequencing and analysis. A.S.B. provided animal-origin influenza virus strains and conceptual input. J.S.Y. conceived the study, supervised the experimental design and performance of experiments, analyzed and interpreted data, and wrote the manuscript.

## Competing interests

The authors declare no competing interests.
