## [Transparent Peer Review file · Nature Communications]

Innate immune control of influenza virus interspecies adaptation via IFITM3

Corresponding Author: Dr Jacob Yount

This manuscript has been previously reviewed at another journal. This document only contains reviewer comments, rebuttal and decision letters for versions considered at Nature Communications.

Version 0:

Reviewer comments:

Reviewer #1

(Remarks to the Author)

Interferon-Induced Transmembrane Protein 3 (IFITM3) has demonstrated strong activity in limiting the reproduction of a variety of enveloped viruses, making it a crucial host antiviral component. Deficits in IFITM3 expression in humans have been found as a risk factor for severe illness after contracting the flu and the SARS-Cov2 virus. The authors from this study extended the prior investigations by analyzing the function of IFITM3 in host susceptibility to infection by a range of influenza A virus and virus adaptability. Although the authors only emphasized the role of IFITM3 in avian and swine influenza virus interspecies adaptation as suggested in the manuscript title, their data supports that IFITM3 has a broad effect on virus infectivity and adaptation regardless of virus host origins. Their study would be more compelling if they included a more extensive NGS analysis of the variants that arose during virus serial passaging in wt and ifitm3 knockout mice. Instead, they only studied the major variants from the H1N1 virus passaging experiment. Overall, their experimental design was reasonable, and analyses were appropriate; however, their study lacks substantial novelty, and Methods section misses certain essential information.

Major and minor comments

1. Fig 1. Three different doses (1, 10, 100 TCID₅₀) of H5N1 and H7N3 virus were used to inoculate wt and ifitm3 knockout mice to determine the minimum infectious dose. However, at 1 TCID₅₀ of inoculation dose, both H5 and H7 virus were able to mount productive infection in all inoculated animals in ifitm3 knockout group, therefore, the minimum infectious dose was not reached. The experiment should include lower inoculation dose (0.1 TCID₅₀) for the ifitm3 knockout group.
2. Fig 2. Considering that A/California/4/2009 and A/Victoria/361/2011 were used in subsequent mouse adaptation experiment, it is a bit surprising that these two viruses were not included in the cell infection assays shown in fig 2. I would assume that IFITM3 has a similar effect on human influenza A virus infection.
3. For cell infection assays, the cells were analyzed for NP expression at 24 hr p.i. There was no description on how infection was done. Was TPCK-trypsin included in the infection medium? Were there single round or multiple rounds of replication during the 24 hr infection period? If authors just want to compare viral infectivity in wt and ifitm3 knockout cells, they should focus on single round infection, which happens around 8 hr post infection.

Fig 2 missed the label for 2f. Also need to include 2f in the figure legend.

4. Line 135. Authors indicate that A/Victoria/H3N2 virus replicated poorly in mice. However, based on the data shown in Fig 3, the wt H3N2 virus reached to above 1×10^4 TCID₅₀/ml in the lung tissues on day 7 p.i, which was less than a log lower than that in the A/California/H1N1 virus group. Was this what they expected from the virus replicated poorly in the mice?
5. Fig 3. NGS data is interesting and helps to identify the mutations emerged during virus serial passage in mice. However, the NGS analysis was only performed for the H1N1 virus. Why the H3N2 viruses from passage experiments were not included? Additionally, for the variants listed in Fig 3 I, the variant frequency needs to be included. There was no mentioning of the threshold for variant calling in the method. Were there more minor variants not included in the fig 3i. Does passaging in ifitm3 knockout mice produce more minor variants?
6. Fig S1: What do the letter/number combinations (such as "MG279989") stand for? Please clarify. Similarly in Table s1, please indicate what the sample IDs stand for. Are they GenBank IDs?

7. Table S1. Human A/California/H1N1 and A/Victoria/H3N2 should be included in this table to validate their assays.
8. Lastly, this might be beyond the scope of this study. The effect of IFITM3 knockout cell lines on the increased viral infectivity shown in fig2 was independent of virus host origins. Have authors considered the effect on viruses with different fusion activation pH? Previous study suggests that the virus with higher fusion threshold may be able to escape IFITM3 restriction by fusing in the early endosomal compartment (PMID: 27707929).

Reviewer #2

(Remarks to the Author)

In this study the authors aim to understand the consequences of IFITM3 deficiency, caused by mutations, on the replication and infectivity of influenza. Through infection of mice with different doses of virus, they find that low viral titres are sufficient to create a replicative infection and induce inflammation in IFITM3^{-/-} mice but not WT. They confirm this in human cell lines using shRNA knockdown. Next, they show that repeated infections in mice can allow mutations to accumulate, providing the virus with an enhanced infectivity, again only in IFITM3^{-/-} mice.

The authors neatly demonstrate the enhanced infectivity of influenza in IFITM3 deficient mice or cells. This result is interesting and potentially explains why the IFITM3 SNPs result in enhanced severity of disease, however more work will be needed to correlate the expression level of IFITM3 in people with these SNPs and infectious load.

The results showing that repeated infections in the absence of IFITM3 can promote virus adaption are also interesting. As the authors state in the discussion, whether this remains true for people with just IFITM3 deficiency rather than absence of IFITM3 would be interesting, but very difficult, to investigate.

Overall, this manuscript provides a new avenue of thought of the impact of IFITM3 deficiency and may provide clues to how SNPs in IFITM3 cause the differences in infection severity for not just influenza, but other viral infections including SARS-CoV-2.

Figure 3c, e: The authors do not discuss the differences in these experiments and the potential reasons for this in the discussion. While variability is expected and I appreciate that they have included the results from 2 independent experiments here, the authors need to address these differences.

Line 137: It is not clear in the text when lung homogenates were taken from the mice during the passages.

Line 139: "Lungs from the infected WT animals were collected at day 7 post infection to measure viral titers as a direct indicator of adaptation to mice when compared to the parental virus isolate." It is not clear after which passage this was done.

Figure S6: This result is valid and should perhaps be moved to a main figure in the paper.

Reviewer #3

(Remarks to the Author)

Reviewer comments

Manuscript: NCOMMS-23-39205-T

Title: Innate immune control of influenza virus interspecies adaptation.

Authors: Denz et al.

Hypothesis: deficiencies in IFITM3 facilitate interspecies infection and adaptation by influenza viruses.

Key Findings

The manuscript confirms previous findings that IFITM3 plays a role in limiting influenza virus replication using a mouse model and standard cells (A549, macrophages, fibroblasts). Passaging of a few viruses in immunodeficient mice/cells resulted in viruses with higher replication efficiency compared to viruses serially passaged in wild-type cells.

Major comments:

1. The association of the study with zoonotic influenza virus infections and pandemics is not clear. The authors used "zoonotic" and "pandemics" in many sentences (e.g. L204-L213) in the manuscript and the title of figures. This can be misleading.

None of the avian-origin viruses in this study were isolated from humans. They were isolated from wild birds. The zoonotic potential of these viruses is extremely low, if at all.

Results in Table S1 showed that H5N1 and H7N3 have no affinity to human-type receptors. They showed the typical affinity to avian-type receptors. Interestingly, the H11N9 virus showed unusual affinity to human-type receptors (Table S1), however, similar to H8, and H14 they have not been isolated from humans and their zoonotic potential are largely unknown/negligible. How do the authors define "zoonotic" viruses?

2. AIV from humans must be used. For example: the recent H3N8, H5N1 (HK/96, VN/04), H7N7/NL/03, H7N9 (Anhui/13), H9N2 (HK/2108/03), a Chinese H10N8. Viruses used are laboratory viruses (PR8, pH1N1) and "1" swine-origin virus. Positive control viruses including more seasonal and pandemic human-origin influenza viruses are missing.

3. The association between IFITM3 deficiency in the human population and the emergence of pandemic viruses is misleading. All pandemic influenza viruses have emerged mainly in Western countries, where the prevalence of IFITM3 SNPs is lower than in China.

4. L237-251: The role of mutations in NP and PB2 on virus replication should be studied by reverse genetics. How can the role of other factors (e.g. defective interfering particles) in virus replication be excluded? It is important to use deep sequencing to understand whether these mutations were present in the quasispecies in P0 and were selected after passaging.

5. L70: Animal experiment:

Body weight results should be provided.

Titration of virus in other organs should be done/shown.

Measurement of IL6 alone is not enough to get an idea of "inflammation", histopathological examination and measurement of type I/II IFNs and other cytokines should be done.

Why the variation in age (6 to 10 weeks) and number (5 or 10) of mice used in the current study?

L129-131, L307, GoF experiment: Protocol # 2016R00000014-R1 for the generation of mouse-adapted viruses should be published. All experiments were performed in BSL2, although they assume increased replication in human cells, and the main arguments in the introduction and discussion are about the zoonotic potential of these viruses.

More information on the animal experiments should be provided: how the inoculum was prepared, replicates of virus titration, back-titration, etc.

6. Replication in cell culture

Titration of infectious virus in A549 and THP-1 should be performed by plaque assay or cell culture (TCID). Detection of NP by flow cytometry is not convincing.

Minor comments:

1. L58: This sentence may be misleading and should be reworded. The emergence of SARS-CoV, SARS-CoV2 and AIV in China has other (mostly cultural) factors. These factors are more convincing than the controversial role of IFITM3 SNPs. The pandemic influenza viruses emerged in countries where IFITM3 SNPs are different.

2. It is important to mention in the introduction that other studies have not found a role for IFITM3 SNPs in influenza.

3. L219: No transmission experiments were performed. This may be misleading.

4. Table S1: the H3N2/Victoria laboratory strain is not listed.

Version 1:

Reviewer comments:

Reviewer #1

(Remarks to the Author)

The authors have fully addressed each of my comments point by point in the revised manuscript, demonstrating careful consideration of the feedback provided. In addition, they have provided additional experimental evidence to further strengthen their conclusions. Overall, I believe that the quality of the manuscript has been improved, and it now presents a more compelling and scientifically sound study.

Reviewer #3

(Remarks to the Author)

Thank you very much for addressing of my comments.

POINT BY POINT RESPONSE TO REVIEWS

Reviewer #1 (Remarks to the Author):

Interferon-Induced Transmembrane Protein 3 (IFITM3) has demonstrated strong activity in limiting the reproduction of a variety of enveloped viruses, making it a crucial host antiviral component. Deficits in IFITM3 expression in humans have been found as a risk factor for severe illness after contracting the flu and the SARS-Cov2 virus. The authors from this study extended the prior investigations by analyzing the function of IFITM3 in host susceptibility to infection by a range of influenza A virus and virus adaptability. Although the authors only emphasized the role of IFITM3 in avian and swine influenza virus interspecies adaptation as suggested in the manuscript title, their data supports that IFITM3 has a broad effect on virus infectivity and adaptation regardless of virus host origins. Their study would be more compelling if they included a more extensive NGS analysis of the variants that arose during virus serial passaging in wt and ifitm3 knockout mice. Instead, they only studied the major variants from the H1N1 virus passaging experiment. Overall, their experimental design was reasonable, and analyses were appropriate; however, their study lacks substantial novelty, and Methods section misses certain essential information.

We have expanded our Methods section as requested, and we also take this opportunity to list three major points of novelty in our manuscript:

Novelty 1. Our results show that IFITM3 deficiency reduces the minimum infectious dose required for viruses to establish an infection *in vivo*. Although it is commonly presumed that innate immunity plays a role in determining the minimum infectious dose of a virus, this fundamental concept lacks concrete evidence in the existing literature. Our study conclusively demonstrates that IFITM3-dependent innate immunity raises the viral dose threshold required for productive influenza virus infection *in vivo*. This is a textbook-level fundamental finding.

Novelty 2. Our results show that IFITM3 deficiency facilitates the adaptation of influenza viruses to a new host species. This finding cannot be inferred from the known functions of IFITM3, particularly since another immunodeficient mouse system with increased virus replication (STAT1 KO) did not provide the same enhancement of virus adaptation.

Novelty 3. Our results identify IFITM3 KO mice as a valuable tool for increasing the rapidity of mouse adaptation of influenza viruses for vaccine and pathogenesis research, as well as for studying the evolutionary processes of interspecies adaptation.

Major and minor comments

1. Fig 1. Three different doses (1, 10, 100 TCID₅₀) of H5N1 and H7N3 virus were used to inoculate wt and ifitm3 knockout mice to determine the minimum infectious dose. However, at 1 TCID₅₀ of inoculation dose, both H5 and H7 virus were able to mount productive infection in all inoculated animals in ifitm3 knockout group, therefore, the minimum infectious dose was not reached. The experiment should include lower inoculation dose (0.1 TCID₅₀) for the ifitm3 knockout group.

Our goal for this work was to determine whether the minimum infectious dose is different in WT versus KO mice. Our results demonstrate that a lower dose of virus can infect IFITM3 KO mice compared to WT mice and, therefore, that the minimum infectious viral dose is lower when IFITM3-dependent antiviral protection is absent. As advised by the editors, further experimentation with lower viral doses was not performed as this would not alter our conclusion.

2. Fig 2. Considering that A/California/4/2009 and A/Victoria/361/2011 were used in subsequent

mouse adaptation experiment, it is a bit surprising that these two viruses were not included in the cell infection assays shown in fig 2. I would assume that IFITM3 has a similar effect on human influenza A virus infection.

We have added A/California/4/2009 and A/Victoria/361/2011 to the cell infection assays in Fig 2 and show that IFITM3 restricts human influenza A virus infections in our cells/assays, consistent with previous studies and the reviewer's expectations.

3. For cell infection assays, the cells were analyzed for NP expression at 24 hr p.i. There was no description on how infection was done. Was TPCK-trypsin included in the infection medium? Were there single round or multiple rounds of replication during the 24 hr infection period? If authors just want to compare viral infectivity in wt and ifitm3 knockout cells, they should focus on single round infection, which happens around 8 hr post infection.

Our results represent a single round of infection as we did not add TPCK-trypsin in the media for these experiments, which used flow cytometry as a readout. For the reasons the reviewer points out, we have often used early 6 h timepoints in experiments on highly susceptible 293T cells and the fast-replicating PR8 virus strain (e.g., Yount, *Nature Chem Biol*, 2010; Hach, *J Virology*, 2013; McMicahel, *JBC*, 2014; Chesarino, *EMBO Reports*, 2017). However, we found that the large panel of diverse human, avian, and swine viruses required longer accumulation of NP to achieve robust detection of infected cells for all viruses. This timing may be due to the non-adapted nature of the animal-origin viruses. Further, we note that the results of our infections are consistent in all cell types, including THP1 macrophages, which generally undergo single cycle abortive replication of influenza viruses. We have added clarifying statements in the results section indicating that these experiments were performed in the absence of trypsin and represent a single cycle infection.

In new experiments, we added TPCK trypsin to A549 cells and measured infectious virus production. We found that IFITM3 deficiency led to increased viral titers for all viruses tested. Thus, IFITM3 deficiency leads to increased initial infection of cells and increased viral output.

Fig 2 missed the label for 2f. Also need to include 2f in the figure legend.

We thank the reviewer for pointing out this omission and have labeled the figure and adjusted the figure legend accordingly.

4. Line 135. Authors indicate that A/Victoria/H3N2 virus replicated poorly in mice. However, based on the data shown in Fig 3, the wt H3N2 virus reached to above 1×10^4 TCID₅₀/ml in the lung tissues on day 7 p.i., which was less than a log lower than that in the A/California/H1N1 virus group. Was this what they expected from the virus replicated poorly in the mice?

We thank the reviewer for pointing out that this statement was not clear. The Victoria H3N2 virus indeed replicates in mice to lung titers of 10^4 . However, it replicates less in comparison to highly mouse-adapted strains, such as the commonly used PR8 or WSN strains, which can reach lung titers of 10^6 - 10^7 particularly when using an inoculum comparable to that used in our experiments. We have thus rewritten the statement that the reviewer refers to. It now reads: "We chose this strain because preliminary experiments indicated that it infects mice, but replicates to lower levels than commonly used mouse-adapted strains, thus providing a virus with significant potential for adaptation to mice."

5. Fig 3. NGS data is interesting and helps to identify the mutations emerged during virus serial

passage in mice. However, the NGS analysis was only performed for the H1N1 virus. Why the H3N2 viruses from passage experiments were not included? Additionally, for the variants listed in Fig 3 I, the variant frequency needs to be included. There was no mentioning of the threshold for variant calling in the method. Were there more minor variants not included in the fig 3i. Does passaging in ifitm3 knockout mice produce more minor variants?

The sequences previously included in Figure 3I were consensus sequences based on variants above 50% frequency. We have now removed this figure and instead include sequencing data for all mouse passaged viruses (Supplementary Tables). These tables show both consensus variants (>50% prevalence) and minor variants (>1%) present for each virus stock. This rich data set shows our previously reported changes in the consensus sequence for the IFITM3 KO-passaged H1N1 virus (largely in polymerase segments) as well as the presence of additional minor variants. The data for H3N2 virus is more complex in that we did not observe consensus sequence changes but did observe emergent minor variants in polymerase segments that likely contribute to the enhanced replication of these stocks. We note, as the reviewer pointed out above, that the H1N1 virus started out with a higher replicative capacity in mice than the H3N2 virus, likely allowing it to more fully adapt in the span of 10 passages. Consistent with replication levels and ability to induce weight loss, the sequencing data suggest that we've captured the H1N1 and H3N2 viruses at different stages of adaptation, though in each case this process occurs more rapidly in IFITM3 KO mice. Virus passaged through STAT1 KO mice also shows emergence of minor variants in polymerase segments that are distinct from the presumed adaptive variants emergent in IFITM3 KOs. We now provide a more thorough discussion of the full set of sequencing data.

Somewhat to our surprise, we do not see an overall increase in variants above the 1% threshold in IFITM3 KO-passaged virus stocks. Rather, in all four of our IFITM3 KO passaging experiments, the stocks generated from the 10th passages in these mice contain the fewest number of minor variants as compared to parental virus and virus passaged in WT mice. This suggests that IFITM3 deficiency may promote viral adaptation by allowing adaptive variants to more efficiently outcompete others in the quasispecies, perhaps by relaxing initial infection bottlenecks as our *in vitro* experiments would suggest.

6. Fig S1: What do the letter/number combinations (such as "MG279989") stand for? Please clarify. Similarly in Table S1, please indicate what the sample IDs stand for. Are they GenBank IDs?

The letter number combinations are the Genbank IDs for the HA gene segments of the individual viruses that were used to construct the phylogenetic table in ED Fig 1. Table S1 has also been updated so that the Genbank IDs are incorporated into the virus nomenclature, such that virus names are listed as they appear in Genbank (e.g., A/black duck/Tennessee/17OS0306/2017 (H5N1)).

7. Table S1. Human A/California/H1N1 and A/Victoria/H3N2 should be included in this table to validate their assays.

The requested data have been added to the table and show a preference of these human viruses for α 2,6 linked sialic acids. Additionally, receptor binding data for the newly examined zoonotic swine viruses have been added to the table.

8. Lastly, this might be beyond the scope of this study. The effect of IFITM3 knockout cell lines on the increased viral infectivity shown in fig2 was independent of virus host origins. Have

authors considered the effect on viruses with different fusion activation pH? Previous study suggests that the virus with higher fusion threshold may be able to escape IFITM3 restriction by fusing in the early endosomal compartment (PMID: 27707929).

While it is beyond the scope of what we investigate here, this would be an interesting avenue to explore in the future. While we note that Sun, et al. found that certain avian viruses partially evade IFITM3 restriction in endothelial cells, influenza virus strains that fully evade IFITM3 activity have not been reported to date. Indeed, the virus studied in Sun, et al. was partially restricted in endothelial cells at low MOIs and robustly restricted by IFITM3 in epithelial cells. We have added a line in our discussion stating that while IFITM3 inhibited infection by all the viruses in all cell types in our experiments, we observed that the degree of inhibition varied depending on the individual viruses, perhaps due to differences in fusion pH optimums leading to differences in fusion in early versus late endosomes as suggested by the cited work.

Reviewer #2 (Remarks to the Author):

In this study the authors aim to understand the consequences of IFITM3 deficiency, caused by mutations, on the replication and infectivity of influenza. Through infection of mice with different doses of virus, they find that low viral titres are sufficient to create a replicative infection and induce inflammation in IFITM3^{-/-} mice but not WT. They confirm this in human cell lines using shRNA knockdown. Next, they show that repeated infections in mice can allow mutations to accumulate, providing the virus with an enhanced infectivity, again only in IFITM3^{-/-} mice.

The authors neatly demonstrate the enhanced infectivity of influenza in IFITM3 deficient mice or cells. This result is interesting and potentially explains why the IFITM3 SNPs result in enhanced severity of disease, however more work will be needed to correlate the expression level of IFITM3 in people with these SNPs and infectious load.

The results showing that repeated infections in the absence of IFITM3 can promote virus adaption are also interesting. As the authors state in the discussion, whether this remains true for people with just IFITM3 deficiency rather than absence of IFITM3 would be interesting, but very difficult, to investigate.

Overall, this manuscript provides a new avenue of thought of the impact of IFITM3 deficiency and may provide clues to how SNPs in IFITM3 cause the differences in infection severity for not just influenza, but other viral infections including SARS-CoV-2.

Figure 3c, e: The authors do not discuss the differences in these experiments and the potential reasons for this in the discussion. While variability is expected and I appreciate that they have included the results from 2 independent experiments here, the authors need to address these differences.

We performed two independent passaging series for both H3N2 and H1N1 because virus adaptation is thought to be a stochastic process involving the random generation and outgrowth of beneficial mutations. In all four of these experiments, we consistently observed that virus passaged through IFITM3 KO mice gained enhanced replicative capacity in WT mice as compared to virus passaged the same number of times through WT mice. The reviewer's careful examination of the H3N2 experiments identifies that the second passage series showed a trend toward adaptation for the virus passaged 10 times through WT mice in terms of virus replication, though this was not to the same magnitude as the IFITM3 KO-passaged virus and was not statistically significant in terms of virus replication. Our newly added sequencing data

for the H3N2 virus stocks identifies minor variants present in the H3N2 series two passages through WT mice that may explain the reviewer's astute observation.

Line 137: It is not clear in the text when lung homogenates were taken from the mice during the passages.

The text of the Results now states that our experimental design used 3 days of infection between passages. The schematic in Fig 3a also visually depicts the 3 day infections between passages and this information is also contained in the figure legend.

Line 139: "Lungs from the infected WT animals were collected at day 7 post infection to measure viral titers as a direct indicator of adaptation to mice when compared to the parental virus isolate." It is not clear after which passage this was done.

Equal doses of passages 0 (parent virus), 1, 5, and 10 were used to infect WT mice for 7 days at which point mice were sacrificed to measure viral titers. To be clear, passaged viruses, whether generated in WT or IFITM3 KO mice, were tested for adaptation in WT animals and lungs were collected at day 7. We have revisited the text of the Results section to add clarifying statements regarding our methods. Our adaptation and testing process is also shown in Fig 3b and described in the figure legend.

Figure S6: This result is valid and should perhaps be moved to a main figure in the paper.

We chose to leave this as a supplementary figure since the STAT1 KO passage series is somewhat tangential to the manuscript's main focus on the function of IFITM3 in viral infection and adaptation. We have, however, added a repeat experiment testing the replication of these passaged viruses in WT mice and now examined the IFN β levels present in the lungs after infection. Intriguingly, we see that while the STAT1 KO-passaged virus is attenuated in terms of virus replication, it induced more IFN β .

Reviewer #3 (Remarks to the Author):

Reviewer comments

Manuscript: NCOMMS-23-39205-T

Title: Innate immune control of influenza virus interspecies adaptation.

Authors: Denz et al.

Hypothesis: deficiencies in IFITM3 facilitate interspecies infection and adaptation by influenza viruses.

Key Findings

The manuscript confirms previous findings that IFITM3 plays a role in limiting influenza virus replication using a mouse model and standard cells (A549, macrophages, fibroblasts).

Passaging of a few viruses in immunodeficient mice/cells resulted in viruses with higher replication efficiency compared to viruses serially passaged in wild-type cells.

Major comments:

1. The association of the study with zoonotic influenza virus infections and pandemics is not clear. The authors used "zoonotic" and "pandemics" in many sentences (e.g. L204-L213) in the manuscript and the title of figures. This can be misleading.

None of the avian-origin viruses in this study were isolated from humans. They were isolated

from wild birds. The zoonotic potential of these viruses is extremely low, if at all. Results in Table S1 showed that H5N1 and H7N3 have no affinity to human-type receptors. They showed the typical affinity to avian-type receptors. Interestingly, the H11N9 virus showed unusual affinity to human-type receptors (Table S1), however, similar to H8, and H14 they have not been isolated from humans and their zoonotic potential are largely unknown/negligible. How do the authors define “zoonotic” viruses?

We have revised our text throughout the manuscript to replace “zoonotic” with the phrases “animal-origin” or “potentially zoonotic” where appropriate. Our results demonstrate that all influenza viruses that we tested were capable of infecting human cells (lung epithelial cells, macrophages, fibroblasts, and cervical epithelial cells), demonstrating that they are potentially zoonotic. Further, our results demonstrate that the unknown zoonotic potential of these viruses is significantly increased by IFITM3 deficiency, which is a major conclusion of our manuscript. Lastly, we have added two confirmed zoonotic influenza viruses to our studies that are documented to have successfully transmitted from swine to humans.

2. AIV from humans must be used. For example: the recent H3N8, H5N1 (HK/96, VN/04), H7N7/NL/03, H7N9 (Anhui/13), H9N2 (HK/2108/03), a Chinese H10N8. Viruses used are laboratory viruses (PR8, pH1N1) and “1” swine-origin virus. Positive control viruses including more seasonal and pandemic human-origin influenza viruses are missing.

We added experiments with the two human viruses used in our passaging experiments (A/California/4/2009 and A/Victoria/361/2011). We also note that we tested 11 avian-origin viruses. The zoonotic viruses that the reviewer suggests require USDA export/import permits and a BSL3+ facility with shower capabilities with which we are not equipped. Instead, as mentioned above, we have added experiments with two additional swine viruses (A/Swine/Ohio/2018 (H1N2) and A/Swine/Ohio/2016 (H3N2)) that are known to have recently infected humans (Sun, *J. Virol.*, 2018; Lorbach, *Mosphere*, 2021). These known zoonotic viruses behave similarly to the human-origin viruses, avian-origin viruses, and our previously used swine-origin virus in terms of ability to infect human cells and enhancement of infection in the absence of IFITM3.

3. The association between IFITM3 deficiency in the human population and the emergence of pandemic viruses is misleading. All pandemic influenza viruses have emerged mainly in Western countries, where the prevalence of IFITM3 SNPs is lower than in China.

As noted above, we removed this statement and reframed the introduction of our manuscript to avoid stigmatizing language. Our work shows that IFITM3 deficiency lowers the initial barrier to infection with animal-origin viruses and promotes interspecies adaptation of influenza virus. Our main conclusions stand independent of the racial distribution of *IFITM3* SNPs.

We have also added discussion of the concept that there is a beneficial tradeoff associated with *IFITM3* SNPs since distinct SNPs resulting in IFITM3 deficiency were positively selected in two independent populations. The prevailing theory in the IFITM field is that IFITM3 deficiency decreases fetal death during certain pregnancy-associated infections. IFITM3 induced by interferons during pregnancy blocks the endogenous retrovirus fusogen-mediated cell-to-cell fusion of placental trophoblasts required for proper placental architecture and function (Buchrieser, *Science*, 2019; Zani, *JBC*, 2019). Thus, human fetuses with IFITM3 deficiencies may have enhanced survival rates during certain infections due to ability to escape the inherent placental toxicity of interferons/IFITM3.

4. L237-251: The role of mutations in NP and PB2 on virus replication should be studied by reverse genetics. How can the role of other factors (e.g. defective interfering particles) in virus replication be excluded? It is important to use deep sequencing to understand whether these mutations were present in the quasispecies in P0 and were selected after passaging.

We have newly tested our adapted virus stocks for replication fitness in human versus mouse lung cells (A549 cells versus LET1 cells). In all cases, we found that the IFITM3 KO-passaged viruses showed increased replication in LET1 cells compared to the parent viruses, but replication in A549 cells was not changed. Understanding the mechanisms by which individual mouse-adaptive mutations increase virus replication specifically in mouse cells is certainly of scientific interest, but this is not the focus of the current manuscript and addition of these data would not change or enhance our conclusions. Additionally, the added sequencing data discussed above that we now provide showing that distinct adaptive mutations emerged in different passaging experiments suggest that the adaptive variants emerged during passaging and their ability to outcompete parental virus may be enhanced in the absence of IFITM3.

5. L70: Animal experiment:

Body weight results should be provided.

Titration of virus in other organs should be done/shown.

Measurement of IL6 alone is not enough to get an idea of "inflammation", histopathological examination and measurement of type I/II IFNs and other cytokines should be done.

Why the variation in age (6 to 10 weeks) and number (5 or 10) of mice used in the current study?

The animals in both H3N2 virus passage series did not lose weight despite increased viral titers seen for the IFITM3 KO-passaged virus stock. We make note of this in the text but do not show figures. Weight loss data for both H1N1 passage series are now shown. We did not collect organs other than lungs in these experiments. We have added lung IFN β ELISA data for each of the *in vivo* experiments as an additional measure of inflammation as requested. As described in the text, groups of 5 mice were used for experiments in which WT mice were challenged with the passaged viruses. For our first H3N2 passage series, we sought to confirm our first results by infecting a second set of mice with the same viruses. Thus, these viruses were tested in 10 mice rather than 5. For reasons regarding mouse deliveries and experimental practicalities, we used mice from 6-10 weeks of age in different experiments though all mice in each individual experiment were age matched. We now make it clear that mice in each experiment were age matched.

L129-131, L307, GoF experiment: Protocol # 2016R00000014-R1 for the generation of mouse-adapted viruses should be published. All experiments were performed in BSL2, although they assume increased replication in human cells, and the main arguments in the introduction and discussion are about the zoonotic potential of these viruses.

The above statement is false and misrepresents our work.

1. We explicitly state that, for biosafety reasons, we chose to adapt human-origin viruses to mice for proof of principle experiments examining whether IFITM3 deficiency affects interspecies adaptation of influenza virus. Mouse adaptation of human-origin viruses has been commonly and safely performed in the influenza field for roughly 100 years. In complete contrast to the reviewer's assertion, mouse adaptation of human-origin viruses is assumed to decrease viral replication/fitness in humans. A prime example of this is the PR8 mouse-adapted strain that caused no detectable replication or immune response in

human clinical trial participants and is thus studied at BSL1 at some institutions due to its enhanced safety profile compared to non-adapted human-origin viruses (Beare, et al., *Lancet*, 1975).

2. We explicitly state in the manuscript that we did not adapt avian viruses to mice (mammals) so as not to increase their zoonotic potential to infect humans.
3. We have added new data confirming that our mouse-adapted H1N1 and H3N2 stocks show increased replication in mouse lung cells (LET1 cells) but do not show increased replication in human lung cells (A549 cells), confirming expectations that our experiments do not provide a GOF for human cell infection.

More information on the animal experiments should be provided: how the inoculum was prepared, replicates of virus titration, back-titration, etc.

We have added the request information to the materials and methods sections to provide more details as requested.

6. Replication in cell culture

Titration of infectious virus in A549 and THP-1 should be performed by plaque assay or cell culture (TCID). Detection of NP by flow cytometry is not convincing.

We examined viral nucleoprotein (NP) via flow cytometry to quantify the percentage of cells that become infected when IFITM3 is present or absent. This directly reports on the outcome of IFITM3 restriction, which blocks infection at the viral entry stage. We show representative flow cytometry plots for each virus infection with all four cell types and demonstrate that the assay is able to easily distinguish non-infected and infected cells. Nonetheless, to address the reviewer's comment, we infected both control and IFITM3 KD A549 cells with a selection of viruses (2 human-origin, 1 swine-origin, 2 swine-origin with known human transmission (zoonotic viruses), and 2 avian-origin viruses). The cells were incubated with TPCK-trypsin to allow for multiple rounds of infection and at 48 hours post infection we collected the supernatants to determine viral titers by TCID50 assay. For all viruses, we see significantly higher titers produced by the IFITM3 KD cells versus control cells. We did not perform this assay in THP1 cells as macrophages generally show abortive replication of influenza viruses and are not likely a major source of viral replication *in vivo*.

Minor comments:

1. L58: This sentence may be misleading and should be reworded. The emergence of SARS-CoV, SARS-CoV2 and AIV in China has other (mostly cultural) factors. These factors are more convincing than the controversial role of IFITM3 SNPs. The pandemic influenza viruses emerged in countries where IFITM3 SNPs are different.

This sentence has been removed. The role of *IFITM3* SNPs is unknown in the emergence of these viruses. Nonetheless, a role for IFITM3 is biologically plausible as IFITM3 is known to restrict each of these viruses. Our present work indicates that cultural factors leading to interactions with viral reservoir species when coupled with IFITM3 deficiencies would enhance the potential for viral zoonosis. We have chosen not to mention cultural factors as these are likely more stigmatizing than our mentions of SNP prevalence.

2. It is important to mention in the introduction that other studies have not found a role for IFITM3 SNPs in influenza.

The available literature indicates that IFITM3 SNPs have the most profound effect in limiting disease caused by emergent viruses (e.g., pH1N1, H7N9, HIV, SARS-CoV-2), suggesting that pre-existing adaptive immunity against seasonal influenza virus strains can largely compensate for IFITM3 deficiency. Studies that have not found a role for *IFITM3* SNPs in influenza usually examined seasonal influenza and/or examined populations in which *IFITM3* SNPs are nearly non-existent, thus making these studies underpowered to make conclusions, which is almost always acknowledged in these manuscripts. We have added a mention of these studies in our introduction.

3. L219: No transmission experiments were performed. This may be misleading.

We now start this paragraph with the following phrase: “While we did not perform virus transmission experiments...” We also state in the next two lines that mice do not transmit influenza virus to each other even when co-housed. Nonetheless, infectious dose is a major factor in viral transmission and we have shown that IFITM3 deficiency reduces the minimal infectious dose of influenza virus *in vivo*.

4. Table S1: the H3N2/Victoria laboratory strain is not listed.

We thank the reviewer for pointing out this omission and both A/California/4/2009 and A/Victoria/361/2011 have been added to Table S1 in addition to the 2 new zoonotic swine-origin viruses we have now examined.